# A New Continuous Flow Microwave Radiation Process Design for Non-Isocyanate Polyurethane (NIPU)

**DOI:** 10.3390/polym15112499

**Published:** 2023-05-29

**Authors:** Ping-Lin Yang, Sung-Han Tsai, Kan-Nan Chen, David Shan-Hill Wong

**Affiliations:** 1Department of Chemical Engineering, National Tsing Hua University, Hsinchu 300044, Taiwan; 2Department of Chemical and Material Engineering, Tamkang University, New Taipei 251301, Taiwan; knchen@mail.tku.edu.tw

**Keywords:** non-isocyanate, polyurethane, microwave, carbon dioxide capture, scale-up, green process

## Abstract

Non-isocyanate Polyurethane (NIPU) has been known to result from a thermal-ring-opening reaction between bis-cyclic carbonate (BCC) compounds and polyamines. BCC can be obtained from carbon dioxide capture using an epoxidized compound. Microwave radiation has been found to be an alternative process to conventional heating for synthesizing NIPU on a laboratory scale. The microwave radiation process is far more efficient (>1000 times faster) than using a conventional heating reactor. Now, a flow tube reactor has been designed for a continuous and recirculating microwave radiation system for scaling up NIPU. Furthermore, the TOE (Turn Over Energy) of the microwave for a lab batch (24.61 g) reactor was 24.38 kJ/g. This decreased to 8.89 kJ/g with an increase in reaction size of up to 300 times with this new continuous microwave radiation system. This proves that synthesizing NIPU with this newly-designed continuous and recirculating microwave radiation process is not only a reliable energy-saving method, but is also convenient for scale-up, making it a green process.

## 1. Introduction

Polyurethane (PU) resins are produced by the polyaddition of polyisocyanates and polyols (see Figure 1) [1,2]. Early in its development, PU was used as a substitute for natural rubber during World War II. The PU structure consists of a hard segment of a carbamate group (-NHCOO-) and a soft segment of long chain polyols. The hard segment is a polar group, whereas the soft segment provides a secondary attraction force through hydrogen bonding and molecular entanglement of the polymer chain, which enhances the performance properties of PU. Its wider applications include apparel, footwear, home appliances, construction, automotive, aerospace, and coatings, etc. [3,4,5,6,7,8,9].

On top of its highly toxic phosgene, polyisocyanate, which is very reactive and dangerous to handle, is an intermediate. Furthermore, handling polyisocyanate raw materials is also a drawback of making traditional PU resin. Therefore, an alternative non-isocyanate process for PU resin has become an attractive topic [10,11].

Kihara et al. (1996) [12,13] published a non-phosgene synthetic route to PU resin using a reaction of epoxy resin with carbon dioxide to cyclic carbonate, then a thermal ring-opening of cyclic carbonate with amines toward PU (Figure 2). This polyurethane preparation does not use isocyanates as an intermediate, so it is called “non-isocyanate polyurethane” (NIPU). One of the handicaps of this NIPU process is that it needs more than 48 h for a thermal ring-opening reaction with diamine, in order to obtain the final NIPU product.

Gedye et al. (1986) [14,15] found that the efficiency of organic synthesis reactions could be improved by using microwave radiation. Its reaction time can be significantly reduced to 1/6~1/240 of the original conventional heating reaction. Since then, the microwave has been extensively used in chemical synthesis. Leadbeater et al. (2002) [16] demonstrated that Suzuki couplings can easily be carried out using water as a solvent in conjunction with microwave heating, with clear advantages of easiness, availability, non-toxicity, and non-flammability. The use of tetrabutylammonium bromide as a phase-transfer catalyst, controlled microwave heating at 150 °C for 5 min, and only 0.4 mol% of Pd(OAc)_2_ used as a catalyst, were found to be key to the success of this method (Figure 3). The same Suzuki couplings [17,18] were also performed under microwave-heated open-vessel reflux conditions and in a preheated oil bath, indicating the absence of specific or non-thermal microwave effects. Moreover, the authors reported another modification in which it was possible to carry out the Suzuki reactions without the palladium catalyst, using TBAB as an additive, Na_2_CO_3_ as a base, and the corresponding boronic acid. Kaval et al. (2004) [19] achieved the Diels–Alder cycloaddition of pyrazinone heterodiene with ethylene, which resulted in the formation of a bicyclic cycloadduct. Under conventional conditions, these cycloadditions had to be carried out in a high-pressure reactor at 25 bar of ethylene pressure and heated for 12 h at 110 °C. Further increasing the temperature did not enhance the reaction rate. At temperatures above 200 °C, a back reaction was observed. Only by using a microwave reactor could the Diels–Alder addition be more efficiently carried out at high temperatures, achieving the reaction in 10 min at 220 °C. Microwave-assisted heating can also be applied to various organic chemistry reactions, including rearrangement reactions [20], heterocycle synthesis reactions [21,22], alkylation reactions [23], oxidation reactions [24], S_N_2 reactions [25,26,27], substitution reactions [28,29,30], and others.

Although most of the studies were performed in small laboratory batch reactors, scale-up and use of flow reactors are possible. For example, Cablewski et al. (1994) [31] developed a flow-through microwave device with the capability of operating at a high pressure (14 bar) and a high temperature (200 °C) for the production of trimethyl benzoate, which is about 1000–1800 times faster than that of the conventional preparation heating method with the same yield results. Correa et al. (1998) [32] found that microwave irradiation can be used to carry out emulsion polymerization reactions in polar solvents more rapidly and conveniently than conventional heating methods. The molecular weight of the polystyrene prepared by microwave irradiation was also higher. However, the risk of explosion is high if overlong periods of time are used, and further research is needed to understand the fundamental effects of microwaves on polymerizations. The study raises important questions about whether microwave energy can alter the mechanism and kinetics of polymerization reactions, which could have significant implications for the polymer manufacturing process. Bogdal et al. (2010) [33] found that the use of phase-transfer catalytic (PTC) conditions in polycondensation reactions resulted in better yields and faster, cleaner reactions, compared to conventional methods. The PTC-catalyzed reactions had the ability to be scaled up to multigram and even kilogram levels of production. Examples of successful PTC protocols included the preparation of poly(etherketone)s, poly(etherimide)s, low-molecular-weight epoxy resins, and polyarylates. PTC protocols offer the advantage of faster and cleaner reactions without any solvents. Muley et al. (2013) [34] published the results of microwave-assisted transesterification of vegetable oil with ethanol for better efficiency, using a scaled-up continuous microwave system. The advantages of microwave heating are very obvious, mainly due to the different sources of heat energy obtained by the reactants. In this study comparing the conversion rates of laboratory-scale mobile systems with those of scaled-up mobile systems, it was found that microwave radiation increases the frequency of intermolecular collisions, accelerating the reaction time and increasing the efficiency of separation and purification due to the increased yield. Energy loss of conventional heating is caused by convection, while microwave radiation provides an electromagnetic field to create molecular rotation and friction among molecules. That generates internal heat. In a comparison between laboratory-scale and scale-up tests of mobile microwave reaction devices, the energy required for the reaction of a unit of vegetable oil in a scaled-up mobile microwave reaction device is only 30% of that required for a laboratory-scale mobile system. It was confirmed that the scaled-up mobile microwave reactor can really improve energy efficiency.

Chen et al. (2013) [35] established that the microwave radiation system can accomplish the ring-opening reaction of Bis-cyclic Carbonate (BCC) with a di-amine on a lab-scale sample within 1 h. The main objective of this work is exploring the NIPU scale-up process with a home-made flow-tube reactor system. Quérette et al. (2019) [36] developed a catalyst-free method for synthesizing linear aliphatic polyurethane without isocyanates, using microwave technology. The synthesis of poly(hydroxy)urethane resulted in the incorporation of pendant hydroxyl groups, primarily secondary ones. The utilization of microwave technology has shown promising results in significantly reducing the reaction time for poly(hydroxy)urethane synthesis. Additionally, the successful preparation of poly(hydroxy)urethane nanoparticles through nanoprecipitation highlights their potential for use in various applications. The choice of solvent, such as DMSO, plays a crucial role in influencing the size and stability of nanoparticles. These findings contribute to the advancement of techniques for synthesizing polyurethane and preparing nanoparticles using microwave assistance. Razali et al. (2022) [37] highlighted the limitations of traditional PU synthesis methods, particularly in terms of time and energy consumption. As an alternative, the authors introduced microwave-assisted synthesis as a more efficient method. They highlighted the promising prospects of microwave-assisted synthesis and the integration of biodegradable components in advancing the production of sustainable PUs. This research contributes to the promotion of greener practices in the polymer industry, fostering a sustainable future. The objective of this work is to explore the feasibility of further scale-up using a commercial batch reactor and self-build recycle flow reactor system.

## 2. Materials and Methods

### 2.1. Materials

Bisphenol A diglycidyl ether (BADGE) with an epoxide equivalent weight (EEW) of 188 is manufactured by Chang Chun Plastics Co. Ltd. in Hsinchu, Taiwan. Ethyl lactate (≥99%), lithium bromide (≥99%), hydrochloric acid (36.5–38.0%), toluene (≥99.5%), and chloroform-D1 (≥99.8%) were procured from Sigma-Aldrich, located in Burlington, MA, USA. Sodium hydroxide (≥97.0%) was purchased from J.T. Baker, Phillipsburg, NJ, USA. Tetrahydrofuran (≥99.9%) was purchased from Duksan Reagents, An-san-si, South Korea. Polyoxypropylenediamine (Jeffamine-D2000) with a molecular weight of 2000 is produced by Huntsman International LLC, located in The Woodlands, TX, USA.

### 2.2. Instrumentation

^1^H NMR spectra were recorded using a Bruker Avance 500 NMR instrument at Bruker Corporation in Billerica, MA, USA, with toluene as the internal standard. FT-IR spectra were recorded on a Thermo Nicolet iS50 FT-IR at Thermo Fisher Scientific Inc. in Waltham, MA, USA. Molecular weight was measured using a Viscotek TDA 305 instrument at Malvern Panalytical in Almelo, The Netherlands. Experiments were conducted using a reaction volume of less than 50 mL in the Discover microwave reactor at CEM Corporation, located in Matthews, NC, USA. The scale-up experiment for the batch process was conducted using a Mars 6 microwave reactor at CEM Corporation, which is situated in Matthews, NC, USA. The recirculating continuous flow process was carried out in a microwave reactor designed by the laboratory.

### 2.3. Preparations

#### 2.3.1. BABCC from Reaction of BADGE with Carbon Dioxide

A homogeneous solution of Bisphenol A diglycidyl ether (BADGE) in ethyl acetate (1500 mL) was prepared by dissolving 2800 g of BADGE (EEW = 188) in the solvent. The solution, which contained 0.5 mole% of lithium bromide as a catalyst, was then mixed in a reactor with a mechanical stirrer. Then, carbon dioxide was introduced into the reaction mixture and maintained at a pressure greater than 1 atm overnight~12 h. A white solid product, BABCC, was collected after centrifugation.

#### 2.3.2. Ring-Opening Reaction of BABCC with Jeffamine-D2000 in Oil Bath

BABCC (4.0 g) and Jeffamine D2000 (20.6 g) were placed in a reaction flask (molar ratio 1.0:1.2) and lithium bromide (0.5 mol%) was added as a catalyst. The reaction was heated in an oil bath with agitation at 130 °C. The reaction was monitored and analyzed by FTIR until the BABCC had disappeared.

#### 2.3.3. Ring-Opening Reaction of BABCC with Jeffamine-D2000 by Microwave Synthesizer

The reaction recipe was the same as the previous one, but it was carried out using the Mars 6 Microwave Synthesizer instead of an oil bath. The reaction was monitored with FTIR.

#### 2.3.4. Batch Reaction of BABCC with Jeffamine-D2000 in a Mars 6 Microwave Synthesizer

BABCC (40.0 g) and Jeffamine D2000 (206.2 g) were placed in a reaction flask (molar ratio 1.0:1.2) and lithium bromide (0.5 mol%) was added as a catalyst. The reaction was irradiated by a Mars 6 microwave reactor at 130 °C. The reaction of BABCC was monitored by FTIR.

#### 2.3.5. Continuous MW Radiation of BABCC with Jeffamine-D2000 in a Recirculating Flow Tube Reactor

Jeffamine-D2000 (6666.67 g) and lithium bromide (15.39 g) were added to the collection tank. The flow rate was set to 4.8 L/min (motor rotation frequency 60 Hz) and the Jeffamine was allowed to circulate through the line. The output power of the microwave was set and the operating temperature was limited to 150 °C, then the microwave reactor was switched on. After heating the reactant in the reactor to 100 °C, 770 g of BCC powder was added to the collection tank at a rate of approximately 2.13 g/min. Heating was performed by microwave radiation for 12 h, with sampling conducted every hour. The conversion rate of BABCC in the samples was analyzed at different reaction times using FTIR.

#### 2.3.6. Continuous Microwave Radiation Flow Tube Reactor

A flow tube reactor has been constructed for continuous recirculating microwave radiation. It is designed for a bigger manufacturing scale of NIPU and its scale depends on the collection tank size (Figure 3). The reaction mixture is fed through the opening (a) above the collection tank and it is pumped into the microwave cavity from the bottom of the cavity using a metering gear pump at a flow rate of 4.8 L/min. The microwave outpower is attenuated at 800~8000 w, based on the celling temperature 150 °C of MW radiated reaction mixture. Radiation from outlet (b) is monitored and the final reaction product is discharged from outlet (c) after the final radiation.

## 3. Results

### 3.1. Analysis of BABCC

#### 3.1.1. FT-IR

The conversion of BADGE to BABCC was confirmed by FTIR spectroscopy. In Figure 4a, BADGE shows an absorption peak at 910 cm^−1^ on the FTIR spectrum, which is indicative of its epoxy functionality. After BADGE reacts with carbon dioxide, the epoxide functional group undergoes a reaction with carbon dioxide to produce a bis-cyclic carbonate. In Figure 4b, the spectrum displays absorption at 1800 cm^−1^, which corresponds to the “C=O” bond stretching vibration signal.

#### 3.1.2. NMR

The purified BABCC in deuterated chloroform (CDCl_3_) was analyzed using ^1^H-NMR spectroscopy on a Bruker Avance 500 NMR instrument (see Figure 5). The ^1^H-NMR spectrum of BADGE prior to the reaction is shown in Figure 5. The hydrogen (H) in epoxy groups a and b of BADGE has signals at 2.8 and 3.3 ppm, respectively. After the epoxide functional group of BADGE reacts with CO_2_, it forms a cyclic carbonate known as BABCC. The chemical shifts of H in a’ and b’ on the cyclic carbonate group of the BABCC are 5.0 and 4.5 ppm.

### 3.2. Analysis of NIPU

#### 3.2.1. Results of FTIR Analysis for Batch System

The conversion of BABCC to NIPU was confirmed through FTIR analysis. In Figure 6a, the FTIR analysis of BABCC before the reaction shows absorption of the cyclic carbonate “C=O” bond at 1800 cm^−1^. Figure 6b shows the FTIR analysis of BABCC after reacting with amines. The signal for the “C=O” bond stretching vibration is at 1720 cm^−1^, while the “N-H” and “C-O” bond stretching vibrations are at 1530 and 1256 cm^−1^, respectively.

#### 3.2.2. Results of FTIR Analysis in Continuous and Recirculating Flow System

The continuous and recirculating microwave radiation system has been separated into two stages: a and b. A represents the reaction mixture of BABCC-Jeffamine in the collection tank. The time from when BCC is added to the end of the reaction is “b”. From Figure 7, it can be observed that at the start of the reaction in the continuous system, there was only Jeffamine D2000 in the reaction system. Therefore, there is no absorption signal of cyclic carbonate and polyurethane. When cyclic carbonate is added to the collection tank at a rate of 10 g/min, it reacts with Jeffamine D2000 to produce polyurethane. As a result, the concentration of cyclic carbonate increases over time. This is because the cyclic carbonate addition rate is greater than the cyclic carbonate reaction consumption rate, as shown by the range of (a) in Figure 7.

When the cyclic carbonate is completely added, its concentration in the reaction system will reach its maximum value. At this point, the area ratio of cyclic carbonate in the FTIR spectrum will also reach its maximum value. Afterward, the cyclic carbonate continues to react with Jeffamine D2000 until all of the cyclic carbonate is consumed. The absorption intensity of polyurethane in the FTIR spectrum increases as the reaction time increases (zone b in Figure 7).

#### 3.2.3. Heating Method

According to Table 1 the NIPU synthesis experiment in an oil bath needs 48 h reaction time, whereas the same reaction size in a Discover Microwave reactor only takes 1 h. The reaction was scaled up by a factor of 10 and completed in only 2 h, using a batch microwave reactor (Discover). In the batch test, productivity increased from 0.51 g/h (using an oil bath) to 24.61 g/h (using Discover). During the continuous microwave radiation scale-up test, the productivity of the Mars 6 was measured at 123.15 g/h. Microwave radiation has been proven to be more efficient than each batch system.

In the continuous flow microwave reactor system, the reaction volume can be increased up to 300 times, and the pump feeds the sample into the microwave cavity at a rate of 3.6 L/min. The total reaction time for the complete reaction between cyclic carbonate and Jeffamine D2000 was 11 h. Productivity increased to 677.46 g/h, which is 27.5 times higher than that of the small batch microwave reaction (Discover).

When the inlet flow rate was increased to 4.8 L/min, the reactant was fully converted to NIPU within 11 h of reaction time. The reaction time remains constant at different feed rates. When the flow rate increases, the reaction flow passes through the microwave cavity at a faster rate, resulting in a shorter exposure time to microwave radiation. As a result, the flow requires multiple exposures to microwave radiation in order to absorb enough energy for the reaction to occur in the microwave reaction system.

#### 3.2.4. Energy Consumption Comparison

To determine if the microwave scale-up process design has an energy-saving effect, compare the total power output of the microwave reactor system during the reaction process. According to Table 2, the turnover energy (TOE) in a small batch microwave reactor (Discover) experiment was 24.38 KJ/g. Due to the small volume of the reactant, a considerable amount of the microwaves are not absorbed, resulting in energy loss. In the experiment using the batch microwave reactor (Mars 6), scaling up the reactant volume by a factor of 10 resulted in a decrease of the total organic carbon (TOC) by 14.86 KJ/g. The power output per unit of time was reduced due to the temperature limitation. However, the heat is not easily dissipated and energy consumption is also reduced. In a continuous microwave reactor, the large volume of the reactant reduces the loss of energy that is not absorbed by the reactant. The temperature of the reactant can be maintained above 100 °C during the external cycle of the microwave cavity. The microwave irradiation within the microwave cavity, combined with the thermal effect from the external circulation, creates a microwave effect. This allows the reactant to continue reacting whether inside or outside the microwave cavity. The process of synthesizing the product requires significantly less energy, only about 8.8 kJ/g, which is one-third of the energy consumed in Discover reactor experiments.

## 4. Conclusions

Microwave radiation has been successfully used for NIPU synthesis in batch sizes of less than 5 g with a reaction time of only 1 h, using various laboratory microwave reactors, such as the Discover and Mars 6 microwave synthesizers. Microwave radiation has been shown to be a highly efficient energy source for NIPU preparation compared to conventional heating methods, such as convection or conduction, which require 48 h. However, NIPU has also been facing the difficulties in scaling up production using microwave reactor (Discover). Here is a report on the design of a tubular reactor design for a continuous microwave radiation process. This new process has demonstrated the preparation of NIPU up to a weight of >7000 gm and its average “Turn Over Energy (TOE)” has been decreased to about one-third of the batch size microwave radiator (24.38 kJ/g). The advantages of a flow-type tubular reactor are manifold. Not only does it provide low TOE with homogeneous NIPU production without batch variations, but it also has the potential to scale up NIPU production through the use of the same continuous tubular-flow process microwave reactor design, enabling future reactor amplification. This continuous microwave radiation is used in the production of NIPU and follows the principles of green chemistry through its continuous process.

## 5. Patents

This study would like to note that some of the findings presented in this paper have been published as patents in China (CN109467664A), Taiwan (TWI663188B), and the United States (US10570254B2).

## Figures and Tables

**Figure 1 polymers-15-02499-f001:**
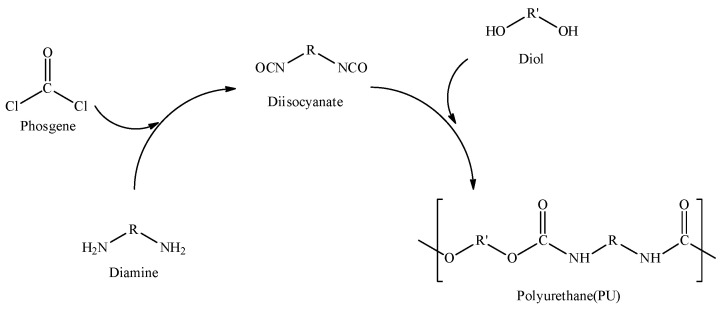
Conventional polyurethane formation.

**Figure 2 polymers-15-02499-f002:**
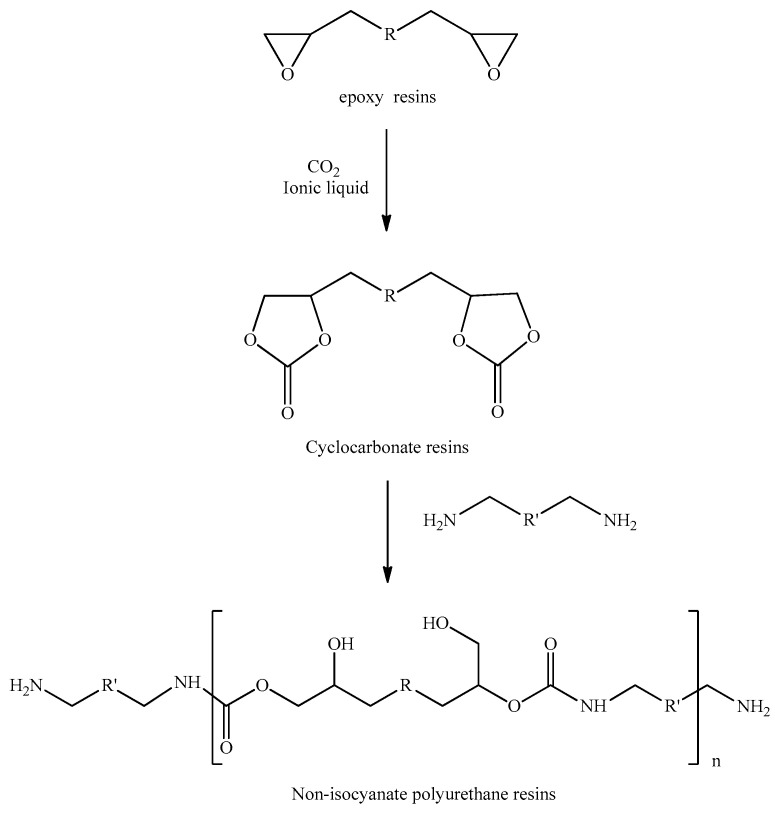
Non-isocyanate polyurethane (NIPU) process.

**Figure 3 polymers-15-02499-f003:**
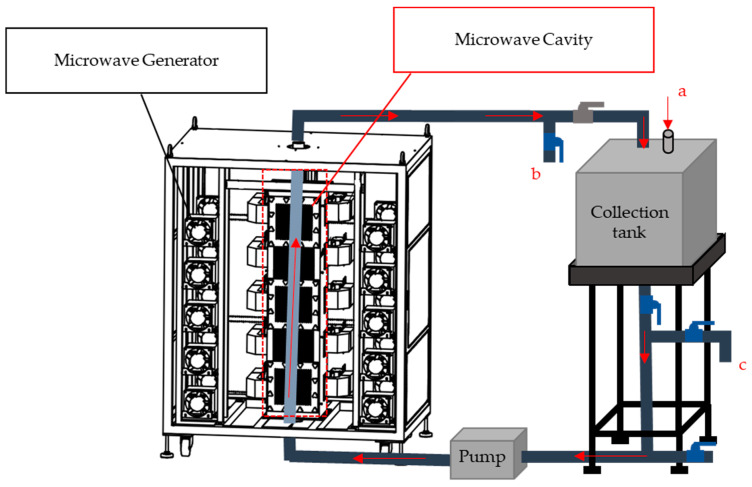
Continuous microwave recirculating flow tube reactor system. (a) Exit for reactant 1, (b) Exit for reactant 2, (c) Exit for reactant 3.

**Figure 4 polymers-15-02499-f004:**
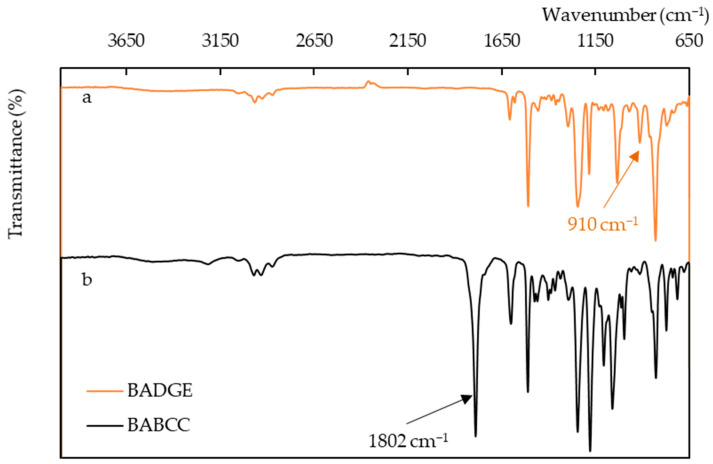
FTIR spectra. (**a**) BADGE, (**b**) BABCC.

**Figure 5 polymers-15-02499-f005:**
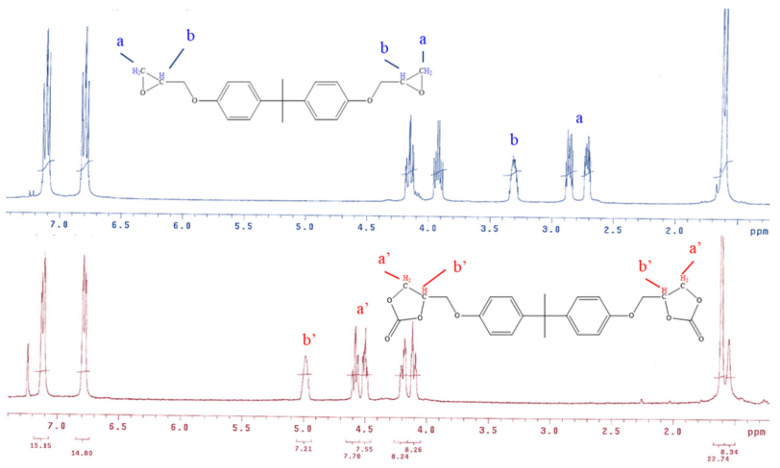
^1^H NMR spectra for BADGE and BABCC in CDCl_3_. Spectrum a and b represent the H-signals of epoxy functional groups a and b of BADGE in ^1^H-NMR. Spectrum a’ and b’ represent the H-signals of cyclic carbonate functional groups a’ and b’ of BABCC in ^1^H-NMR.

**Figure 6 polymers-15-02499-f006:**
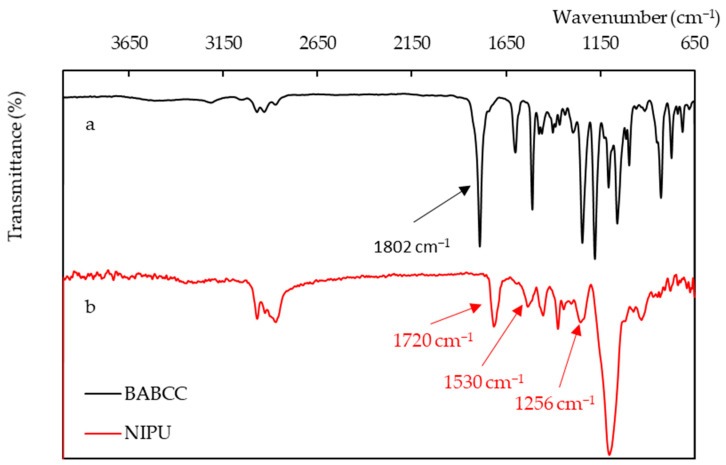
FTIR spectra. (**a**) BABCC, (**b**) NIPU.

**Figure 7 polymers-15-02499-f007:**
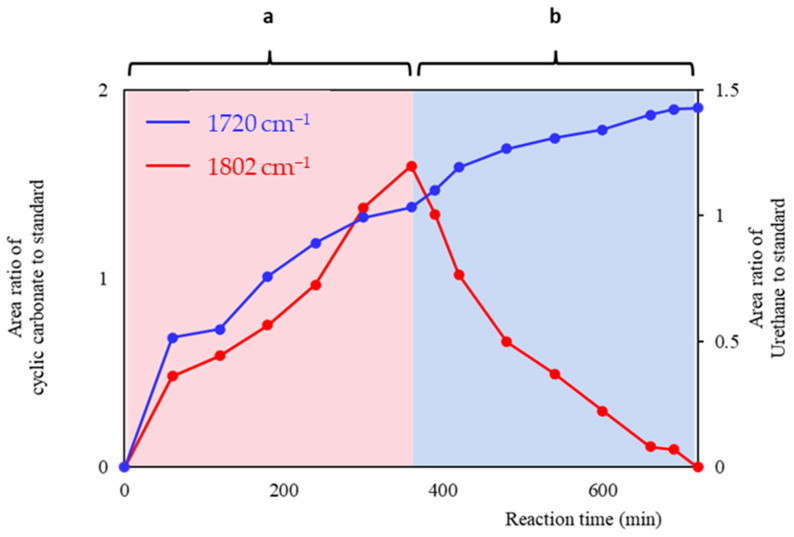
Absorption intensity of cyclic carbonate/urethane products under continuous microwave radiation as a function of time in FTIR. (**a**) represents the time interval for starting to add BCC. (**b**) represents the time interval between the addition of all BCC and the end of the reaction.

**Table 1 polymers-15-02499-t001:** Productivity of experiments in different reaction systems.

	Reaction Condition	Reactants (g)	Time (h)	Temperature (°C)	Flow Rate (L/min)	Productivity (g/h)
entry 1	Oil bath	24.61	48	130	-	0.51
entry 2	Discover	24.61	1	130	-	24.6
entry 3	Mars 6	246.30	2	130	-	123
entry 4	Continuous flow system	7452.06	11	125–145	3.6	677
entry 5	Continuous flow system	7452.06	11	125–145	4.8	677

**Table 2 polymers-15-02499-t002:** Energy consumption of microwave system process amplification.

	Reaction Condition	Reactants (g)	Time (h)	Flow Rate (L/min)	Total Energy Consumption (kJ)	Turn over Energy ^c^ (kJ/g)
entry 2	Discover	24.61	1	-	600 ^a^	24.38
entry 3	Mars 6	246.30	2	-	3660 ^a^	14.86
entry 4	Continuous flow system	7452.06	11	3.6	66,254 ^b^	8.89
entry 5	Continuous flow system	7452.06	11	4.8	65,427 ^b^	8.78

Total energy consumption is calculated from the output power and time. ^a^
$\overset{t}{\underset{t=0}{\sum }}\frac{W_{t}\times \Delta t}{1000}$, *Wt*: the power provided by the batch microwave reactor per second, ∆*t*: interval of time. ^b^ $\overset{t}{\underset{t=0}{\sum }}\frac{W_{t}\times \Delta t}{1000},\;$*Wt*: the power provided by the continuous flow system microwave reactor per minutes, ∆*t*: interval of time. ^c^ Turn over energy = total energy consumption (kJ)/amount of reactant.

## Data Availability

All data generated during this study are included in the article.

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
