# Peer review of "A New Continuous Flow Microwave Radiation Process Design for Non-Isocyanate Polyurethane (NIPU)"

_polymers, 2023, doi:10.3390/polym15112499_

Round 1

Reviewer 1 Report

This manuscript, entitled “A New Continuous Flow Microwave Radiation Process Design for Non-Isocyanate Polyurethane (NIPU)”, describes the preparation of non-isocyanate polyurethanes using microwave radiation in the laboratory scale as well as in the scale-up scope. In addition, this manuscript provides a comparison of heating methods and energy consumption when performing reactions with different equivalents of reactants. It presents a new alternative synthetic method for the preparation of non-isocyanate polyurethanes. It can be published after major revision.

1. The literature on microwave radiation technology in the introduction is old and needs to be expanded with literature on microwave radiation technology from the last five years.

2. The manuscript contains some spelling errors and needs further revision and polishing in terms of professionalism and rigor of language presentation.

3. The headings in the IR spectra in the Discussion of Results section are inappropriately positioned and the font size is not uniform; the clarity and standardization of the IR spectra need further revision.

4. The clarity and resolution of the 1H-NMR are too low and no specific attribution information is given on the 1H-NMR. In addition, the corresponding 13C-NMR need to be given and the solvent used indicated.

5. In the analysis of the IR spectrum of BABCC, the IR absorption peak at 1800 cm-1 is attributed to “C=O” rather than “C-O”. The manuscript should be revised for correctness and rigour in the analysis of the results.

6. When the manuscript compares the energy consumption of microwave radiation systems, the text does not correspond to the values in Table 2, e.g. “turn over energy (TOE) was 24.38 KJ/g” in the text, but no relevant data were found in Table 2.

7. The specific calculation of Total energy consumption and Turn over energy in this paper should be given in detail in the text.

The manuscript contains some spelling errors and needs further revision and polishing in terms of professionalism and rigor of language presentation.

Author Response

Dear Reviewer,

We would like to express our gratitude for your review of our manuscript titled "A New Continuous Flow Microwave Radiation Process Design for Non-Isocyanate Polyurethane (NIPU)" and for providing us with valuable feedback. We have carefully considered your comments and suggestions and appreciate the opportunity to address them in this revised version.

  1. The literature on microwave radiation technology in the introduction is old and needs to be expanded with literature on microwave radiation technology from the last five years.

We acknowledge your comment regarding the need to expand the literature on microwave radiation technology in the introduction, particularly with recent publications from the last five years. In the revised manuscript, we have updated our literature review by including significant studies published in 2019 and 2022 that focus on the use of microwave radiation in polyurethane synthesis. These additional references will provide a more comprehensive overview of recent advancements in the field.

  1. The manuscript contains some spelling errors and needs further revision and polishing in terms of professionalism and rigor of language presentation.

We apologize for any spelling errors and lack of professionalism or rigor in the language used. We will conduct a comprehensive revision to ensure the manuscript's clarity, accuracy, and overall linguistic quality.

  1. The headings in the IR spectra in the Discussion of Results section are inappropriately positioned and the font size is not uniform; the clarity and standardization of the IR spectra need further revision.

Thank you for pointing out the inappropriate positioning and inconsistent font size of the headings in the IR spectra section. We will reorganize the headings and ensure the uniformity of the font size. Additionally, we will carefully review and standardize the IR spectra to enhance their clarity and improve their presentation.

  1. The clarity and resolution of the 1H-NMR are too low and no specific attribution information is given on the 1H-NMR. In addition, the corresponding 13C-NMR need to be given and the solvent used indicated.

We appreciate your comment regarding the clarity and resolution of the 1H-NMR spectra. In the revised manuscript, we will provide higher resolution figures of the 1H-NMR spectra, along with detailed attribution information. We would like to clarify that the solvent used for the NMR analysis was CDCl3. We apologize for any confusion caused by our previous response, which mentioned 13C-NMR. We will ensure that the revised manuscript accurately reflects the use of 1H-NMR spectroscopy for the analysis of our samples. Furthermore, considering the resolution issue in the NMR spectra, we have decided to retest the samples to obtain better quality data. However, due to ongoing maintenance of the NMR instrument at our institution, we kindly request a two-week extension to properly address this matter.

  1. In the analysis of the IR spectrum of BABCC, the IR absorption peak at 1800 cm-1 is attributed to “C=O” rather than “C-O”. The manuscript should be revised for correctness and rigour in the analysis of the results.

We apologize for the incorrect attribution in the analysis of the IR spectrum of BABCC. We have revised the manuscript to rectify this mistake and ensure the accuracy and rigor of our analysis of the results.

  1. When the manuscript compares the energy consumption of microwave radiation systems, the text does not correspond to the values in Table 2, e.g. “turn over energy (TOE) was 24.38 KJ/g” in the text, but no relevant data were found in Table 2.

We acknowledge the discrepancy between the text and the values presented in Table 2 regarding the energy consumption of microwave radiation systems. We have carefully revised the text to ensure it accurately corresponds with the data provided in the table, addressing the mentioned discrepancy.

  1. The specific calculation of Total energy consumption and Turn over energy in this paper should be given in detail in the text.

We appreciate your suggestion to include a detailed calculation of Total energy consumption and Turn over energy in the text. In the revised manuscript, we have included specific calculations to ensure transparency and clarity in presenting energy consumption data.

We sincerely apologize for any confusion caused by our previous mistakes and appreciate your understanding. We are confident that the revised manuscript, with the necessary corrections and improvements, will provide a more comprehensive and accurate account of our research findings.

Thank you once again for taking the time to provide your valuable feedback. We look forward to submitting the revised manuscript again and receiving further guidance.

Best regards,

Ping-Lin Yang

Reviewer 2 Report

The manuscript "A New Continuous Flow Microwave Radiation Process Design 2 for Non-Isocyanate Polyurethane (NIPU)" is good to read however, all the studies are too old. No recent reference has been given and no current research has been compared before performing the work. It is surprising to see that there are no references between 2013 to 2023. Scarcely 1 or 2 which is totally unexpectable. The manuscript should have been published years before.

English needs to be improved a lot by professionals.

Author Response

Dear Reviewer,

Thank you for reviewing our manuscript titled "A New Continuous Flow Microwave Radiation Process Design for Non-Isocyanate Polyurethane (NIPU)." We appreciate your time and valuable feedback on our work. We have carefully reviewed your comments and made the necessary revisions to address your concerns.

We acknowledge your concern regarding the absence of recent references and comparisons with current research in the field. In the revised manuscript, we have extensively updated the literature review section to include recent references, with a specific focus on microwave applications in polyurethane synthesis that were published in 2019 and 2022. We have also compared our work with the latest advancements in the field, providing a comprehensive overview of the current state of research.

We apologize for the omission of references from 2013 to 2023. We recognize the significance of incorporating up-to-date research in our study, and we have addressed this issue by including pertinent references from the specified timeframe. The revised manuscript now offers a more current and thorough analysis of the available literature.

We acknowledge your comment regarding the timing of our manuscript's publication. We apologize for any misunderstandings regarding the timeline for publication. Our intention is to make a meaningful and relevant contribution to the field of research. We believe that the revised manuscript, which includes recent references and comparisons, will be a valuable addition to the existing body of knowledge.

Once again, we sincerely appreciate your valuable feedback. It has helped us improve the quality and relevance of our work. We hope that the revised manuscript now meets your expectations and has addressed the concerns that were raised. We look forward to submitting the revised manuscript again and receiving additional guidance.

Thank you for taking the time to consider this.

Best regards,

Ping-Lin Yang

Reviewer 3 Report

This report introduces a tubular reactor design for continuous microwave radiation processes, which has successfully demonstrated non-isocyanate Polyurethane (NIPU) preparation in sizes up to >7000 gm. The product was convincingly characterized. The process has also been demonstrated to reduce energy consumption to about 1/3 of that of the batch microwave radiator. The flow-type tubular reactor also shows an advantage in homogeneous NIPU production without batch variations with great potential to scale up NIPU production for green chemistry. This research is meaningful for both the academic and industrial communities. The manuscript is well-written, and experimental data well support the conclusions. Based on these significances, I recommend the publication of this paper on Polymers after a minor revision. Some typos exist. The quality of Figure 5 needs to be improved.

Some typos errors are still present, for example, in Figure 3, "Micrewave Generator"

Author Response

Dear Reviewer,

Thank you for reviewing our manuscript titled "A New Continuous Flow Microwave Radiation Process Design for Non-Isocyanate Polyurethane (NIPU)" and providing us with valuable feedback. We appreciate your positive evaluation of our work and your recognition of its significance for both the academic and industrial communities. We have carefully reviewed your comments and made the necessary revisions to address the minor issues you raised.

We have thoroughly proofread the manuscript and corrected any typos and errors to ensure the clarity and professionalism of the content. We apologize for any oversights in the initial submission and appreciate your diligence in bringing them to our attention.

We acknowledge your comment regarding the quality of Figure 5, which presents the NMR spectra. After reviewing the figure, we have taken steps to enhance its quality. However, we encountered technical difficulties in obtaining the desired resolution for the NMR spectra. Therefore, we have decided to retest the samples in order to obtain better quality data. Unfortunately, our NMR instrument is currently undergoing maintenance, and we anticipate a delay of 1-2 weeks before we can access it. We kindly request your understanding and flexibility in granting us additional time to address the issue with the NMR spectra.

Once again, we sincerely appreciate your positive feedback and recognition of the significance and quality of our work. We believe that the revisions we have made have further strengthened the manuscript and addressed the minor issues that were raised. We are grateful for your guidance and support throughout the review process.

We are eagerly anticipating the opportunity to resubmit the revised manuscript to Polymers and are optimistic that it will be accepted for publication. Thank you for dedicating your time, expertise, and making a valuable contribution to the enhancement of our research.

Best regards,

Ping-Lin Yang